# Lamotrigine Therapy: Relation Between Treatment of Bipolar Affective Disorder and Incidence of Stevens–Johnson Syndrome—A Narrative Review of the Existing Literature

**DOI:** 10.3390/jcm14124103

**Published:** 2025-06-10

**Authors:** Kacper Żełabowski, Kacper Wojtysiak, Zuzanna Ratka, Kamil Biedka, Agnieszka Chłopaś-Konowałek

**Affiliations:** 1Student Society for Psychopharmacology and Forensic Toxicology, Wroclaw Medical University, 4 J. Mikulicza-Radeckiego Street, 50345 Wroclaw, Poland; kacper.zelabowski@gmail.com (K.Ż.); kacper.wojtysiak@student.umw.edu.pl (K.W.); zuzanna.ratka@student.umw.edu.pl (Z.R.); 2Department of Physiology and Pathophysiology, Division of Pathophysiology, Wroclaw Medical University, Chalubinskiego 10, 50368 Wroclaw, Poland; kamil.biedka@umw.edu.pl; 3Department of Forensic Medicine, Division of Molecular Techniques, Faculty of Medicine, Wroclaw Medical University, Sklodowskiej-Curie 52, 50369 Wroclaw, Poland

**Keywords:** bipolar affective disorder, lamotrigine, Stevens–Johnson syndrome, mood stabilizers, hypersensitivity reactions, cutaneous adverse drug reactions

## Abstract

Lamotrigine is the drug of choice for the treatment of depressive episodes in bipolar disorder (BD). Despite its generally favorable tolerability profile, lamotrigine use is associated with a risk of Cutaneous Adverse Drug Reactions (cADRs), including Stevens–Johnson Syndrome (SJS) and Lyell’s syndrome, also known as toxic epidermal necrolysis (TEN). Genetic markers HLA and, in particular, HLA-B 15:02 and HLA-A 31:01 are crucial in predicting individuals’ susceptibility to developing the symptoms. The symptoms are triggered by type IV hypersensitivity developing because of CTL and NK cell activation, leading to keratinocyte apoptosis, epidermal necrosis and skin detachment. The exact pharmacotherapy that should be widely utilized in treating affected patients has not yet been established. New therapies including JAK inhibitors or cyclosporine show potential in improving outcomes by reducing mortality and enhancing the period of recovery. Key factors in preventing cADRs may include adequate patient observation, gradual titration of the patient’s dose, and reduction of risk factors through screening for HLA polymorphisms. When the initial symptoms of cADR are identified, it is imperative to make an immediate decision to discontinue treatment, as this can significantly reduce the risk of progression to SJS/TEN and systemic complications. The purpose of this review is to identify a significant correlation between lamotrigine use in BD and the occurrence of SJS by showing the risk factors, neuropharmacological mechanisms, immune response and correctness of pharmacotherapy.

## 1. Introduction

### What Is Bipolar Affective Disorder and Its Potential Hazards?

Bipolar disorder (BD) affects approximately 60 million people worldwide and its global prevalence is estimated at 2.4%. BD is a chronic mental disorder with repetitive mania/hypomania as well as depressive episodes, which results in marked impairment in overall functioning and deterioration of the patient’s quality of life [1]. The unclear and not fully defined pathogenesis of the disease determines the clinical diversity of patients. A dissimilarity exists between Bipolar Disorder Type I (characterized by its manic episodes) and Bipolar Disorder Type II (characterized by cyclical hypomania and depression). The genetic underpinnings of these conditions are partially shared with other mental disorders. Type I disorder exhibits a stronger genetic correlation with schizophrenia compared to Type II disorder, while Type II disorder demonstrates a stronger correlation with depressive disorders compared to Type I disorder [1].

A proclivity for a self-destructive lifestyle has been documented to reduce life expectancy by 10 to 20 years in affected individuals [2]. Patients with bipolar disorder show a higher chance of morbidity from metabolic and cardiovascular disease entities [3]. Reduced life expectancy has been demonstrated to be associated with successful suicide attempts. According to statistical data, 0.9% of individuals diagnosed with BD die as a result of a completed suicide attempt, which is equivalent to 0.014% of the general population [3]. Bipolar disorder has the highest suicide rate of all mental illnesses, with rates of suicide being increased by up to 15 to 30 times the global population trend [4]. There are several factors that increase the risk of a suicide attempt. These include an autoimmune incident in the family, male gender, divorce, no offspring, and Caucasian race [5]. Such knowledge of family medical history is an important factor in diagnosing the disorder. It is worth noting that suicide is associated with depressive or mixed states, not mania [3].

The treatment of bipolar affective disorder and the implementation of appropriate treatment procedures are hampered because about 75% of the first symptoms of the disease begin with a depressive state [6]. This results in a misdiagnosis of major depression disorder (MDD) affecting 40% of patients currently treated for BD-related disorders. An additional factor that hinders the efficacy of pharmacotherapy is the failure of patients to adhere to the proper medical advice. More than 50% of patients diagnosed with a particular condition did not properly take the medications prescribed by their doctor [4,7]. The provision of adequate psychological support from medical facilities is an integral component of comprehensive therapeutic care [8]. The synergy of pharmacotherapy and comprehensive mental healthcare facilitates the implementation of a holistic therapeutic model.

## 2. Lamotrigine—A New Option in the Prevention of Depressive Episodes

Lamotrigine (LTG) is a fundamental therapeutic agent in the management of bipolar affective disorder. It is classified as a mood-stabilizing agent, which, in conjunction with lithium and valproic acid, facilitates the mitigation of the symptoms associated with cyclic episodes of mania and hypomania, as well as depression. Among the entire group of stabilizers, lamotrigine exhibits the most pronounced antidepressant effect. Lamotrigine is often referred to as a “mood stabilizer from below” because, with long-term therapy, it alleviates the symptoms of chronic depressive episodes and also benefits patients with comorbid anxiety disorders. In the context of classifying the oscillation period of mood changes in BD, lamotrigine demonstrates optimal efficacy during the rapid phase change type [5].

Lamotrigine, however, exhibits reduced efficacy in manic episodes relative to lithium. Greil et al. [9] presented data on the decline in lithium prescriptions by physicians to patients with BD, which stems from physicians’ belief that with current treatment alternatives, the side effects of lithium outweigh the potential benefits of treatment. The prevalence of adverse effects may be lower than previously estimated, and alternative medications such as olanzapine, quetiapine (atypical antipsychotics with mood-stabilizing properties), and other stabilizers may carry a spectrum of additional side effects [9]. These data are particularly relevant in the context that lithium is the most effective antisuicidal stabilizer [10]. Overall efficacy in the treatment of BD is thought to be identical between lamotrigine and lithium [6].

Administration of lamotrigine is associated with a number of adverse effects, including headaches, dizziness, ataxia, and mild rash. Some of the most serious complications, which may be fatal, include Stevens–Johnson syndrome (SJS), toxic epidermal necrolysis (TEN, also known as Lyell’s syndrome) and Drug Reaction with Eosinophilia and Systemic Symptoms (DRESS). The etiology of these disease entities is classified as severe cADR [6,7]. SJS and TEN share a common pathogenesis, and their differentiation is based primarily on the percentage of body surface area (BSA) occupied [11]. Lamotrigine has been identified as the second most likely pharmaceutical agent to induce DRESS after allopurinol. Moreover, it has been documented as the fifth-most drug with a statistically significant correlation between its administration and mortality following this specific adverse effect [12].

More than half of patients taking lamotrigine experience Cardiac Conduction Delays (CCDs) [13]. Disorders related to cardiac function justify the fact that the American Epilepsy Society (AES) issued a recommendation in January 2021 indicating the performance of an ECG in people over the age of 60 who start treatment with lamotrigine. Also included in the group of people with an indication for an ECG are those under the age of 60 with known heart disease or at risk of cardiovascular disease (CVD) [13]. Meyer et al. [14] presented in a cohort study that carbamazepine and valproic acid are associated with an elevated risk of cardiovascular events in comparison to lamotrigine [14].

In addition to the suicides mentioned earlier, equally important side effects include atrial septal defects, Brugada Syndrome and signs suggestive of aortic stenosis [15]. In order to minimize the likelihood of adverse effects, it is advisable to initiate lamotrigine dosing with relatively smaller amounts, which should be consistently increased over time until the desired dose is reached [7,8], which is described in detail in Section 4 (Dosage, Serum Concentration and Therapeutic Efficacy).

### 2.1. Mechanism of Action of LTG

The anticonvulsant action of LTG is typically attributed to its voltage-gated sodium channel (Nav channel) blocking action, which does not really distinguish it from other drugs from this class. Inhibition of Na channels leads to dampening of depolarization rates in CNS and consequently to decreased glutamate and aspartate release from presynaptic membranes (which are themselves stimulating neurotransmitters) [16]. Keeping in mind that excessive synaptic transmissions caused in part by glutamate are at the core of epilepsy pathogenesis, rationale for usage of LTG in anticonvulsant therapies is straightforward. Potentiation of GABA signaling is also postulated to be present and having some influence [17,18], but inhibition as well as induction of this neurotransmitter pathway have been reported [19].

As LTG is also used for maintenance therapy, such chronic exposure and consequently general inhibition of neuronal transmission could possibly have a negative impact on physiological and cognitive processes during states in-between convulsions and make the drug dangerous. This is not an issue, however, because of the exact mechanism of Na channel binding by LTG. The channels cycle between inactivated and activated states, the first one being the most probable conformation for a channel in a hyperactive and hyper-responsive neuron [19]. By binding specifically to the inactivated form, LTG (and other same-class drugs like phenytoin) depresses only pathological, hyperactive signals, and day-to-day cognitive impairments are not being reported [18,20]. Modulation of other channels’ function, such as N-, P-, and Q-type calcium channels, and influence on H+ current are possible secondary actions responsible for LTG’s therapeutic effect, although the effect on sodium current is the most significant. This may also explain why some types of seizures do not respond as readily to LTG as to other agents (seizures whose pathogeneses rely mainly on ions other than sodium) [18,19]. Even though LTG’s receptor profile is diverse, it is not broad enough to be on par with convulsion-type diversity. Nevertheless, its superiority in clinical versatility in comparison with drugs like ethosuximide is undeniable.

Although, to date, we lack indisputable evidence on the mechanism of LTG’s euthymic (mood-stabilizing) action (despite excessive research that takes place to this day), dampening of a broad selection of intracellular signaling pathways seems like a rational explanation [17]. LTG decreases the overall activity in the anterior cingulate cortex and bilateral amygdala, which suggests modulation of emotion handling by the brain. An idea that puts LTG’s glutamate-inhibiting action into the spotlight can be backed up by studies in which the activity of glutamate in the cells of the dentate gyrus was reduced (preclinical model). The influence on glutamate signaling may also a reason for LTG’s antimanic action and is a reason for its interaction with ketamine [21,22]. Furthermore, on an animal depression model, veratrine (a Na channel-opening agent) exhibited antagonistic effects to LTG’s antidepressant effect. Serotoninergic and noradrenergic system regulations were also suggested to be involved [21,22]. It would be reasonable to conclude that LTG’s euthymic action basis lies solely in the inhibition of negative emotion pathways or at least the blockage of their hyper-responsiveness. However, studies on brain electrical activity during exposure to LTG suggest that all processes involved in emotion processing are impaired regardless of the type of emotion [22].

Apart from the anticonvulsant and euthymic actions of LTG, the importance of which is already set in stone, possible new uses may emerge in the near future. Hyper-responsiveness and disturbances of sleep, which are present to some degree on the autism spectrum, can in some cases be attributed to dysregulation of HCN2 channels in brain tissue [23,24]. In animal studies by Guo et al., LTG has been shown to mitigate these effects by modulating the function of said channels [24]. Another study involving children confirmed that in 8 out of 13 participants, negative sensory effects associated with autism were improved [24]. HCN2’s involvement in the fundamental neural process known as synaptic plasticity, which is said to be crucial for learning, memory and other cognitive processes, may be the reason why patients on LTG report improved quality of life and overall improved psychological well-being [18]. An action not related to ion channel modulation that is worth noting is the neuroprotection that is likely caused by down-regulation of pathways responsible for Reactive Oxygen Species (ROS) production in cells and consequent decreases in oxidative damage and neuronal death. Dampening of glutamate signaling by itself may promote neural cell protection by decreasing excitotoxicity (a well-known neurobiological effect of glutamate) [18,21,25]. LTG’s favorable effects on concentrations of malonodialdehyde, NO, rotenone and 1-methyl-4-phenylpyridinium (which are inversely proportional to GSH efficiency in radical scavenging) support its contribution to the beneficial redox state of the cells, leading to better neuron survivability. Furthermore, signaling through the NMDA receptor (for which glutamate is the substrate) may be a basis for the neuroprotective action of LTG, as it increases calcium current and subsequently excessive intracellular calcium levels [26]. A simplified diagram of the mechanisms of LTG’s action is presented in Figure 1.

### 2.2. Pharmacokinetics of Lamotrigine

Lamotrigine is rapidly and entirely absorbed after oral administration with minimal first-pass metabolism effects. The bioavailability of LTG exceeds 98% and is not affected by food [27]. Maximum plasma concentrations occur approximately 2.5 h after oral administration of lamotrigine. Plasma protein binding is approximately 55%. The volume of lamotrigine distribution ranges from 0.92 to 1.22 L/kg, slightly increasing in patients with epilepsy to a range of 1.28 to 1.36 L/kg [27]. It is extensively metabolized in the liver by the action of uridine diphosphate glucuronosyltransferases (UGTs), mainly UGT1A4 and UGT2B7 isoforms, to inactive metabolites 2-N-glucuronide conjugate, 5-N-glucuronide and a 2-N-methyl metabolite [27]. Less than 10% of the drug is excreted unchanged in the urine. Only about 2% of lamotrigine derivatives are excreted in the feces. The pharmacokinetics of LTG show a high interindividual variability which can be caused by several factors such as weight, sex, race, age, comedication, pregnancy, oral contraceptives, creatinine clearance, smoking and genetic variants. Younger children (0.17 to 5 years) eliminate lamotrigine faster than older children (5 to 10 years). Children may be more prone to enzyme induction than adults. The mean plasma half-life in healthy subjects is estimated to be approximately 33 h (range 14 to 103 h). The mean half-life of lamotrigine may be shortened to approximately 8–14 h when administered with carbamazepine and phenytoin, which, like lamotrigine, induce glucuronidation. Valproate, also an antiepileptic/euthymic, is a potent inhibitor of many CYPs and UGTs, increasing LTG’s half-life to as much as 60 h—roughly threefold the base value [18,28,29]. Therefore, it is worth mentioning that some alternative co-agents have a negligible impact on LTG clearance, like levetiracetam, as this combination minimizes pharmacokinetic risk [18]. Apart from drugs, hormones like estrogens have an inducing effect on UGT expression in the liver [18]. Thus, in pregnancy and chronic OC (oral contraceptive) usage, lamotrigine’s level can be disturbed by a substantial amount and precipitate seizures or at least mitigate its therapeutic action. In such patients, it is necessary to assess their drug level frequently as the third-trimester lamotrigine level is, on average, 42% of the baseline. Thus, taking into account LTG’s somewhat narrow therapeutic reference range, doses should be escalated following the first trimester of pregnancy, especially in high-risk women [18,30]. As for contraception, even though studies have mainly focused on oral agents [31,32], it is advised to inform patients who prefer intravaginal devices or transdermal patches about possible interactions with lamotrigine, as systemic effects of a non-systemically oriented therapy may not be as easy to deduce [33]. It is also worth noting that age (which, we presume, is in part because of progressive liver enzyme maturation), body weight and liver efficiency can also have an influence on LTG pharmacokinetics [28]. Another interesting finding related to UGTs’ versatile function, in this case, lipid metabolism influence, as outlined by Nerbert and Russel [34], is of clinical value. Actions that lead to mobilization of adipose tissue storages, like the popular ketogenic diet, were found to decrease the concentration of some agents, including LTG, although other drugs were affected more. This knowledge may be useful when assessing therapy for patients of great risk [35]. On top of that, as a conclusion to animal model studies, when prescribing LTG, it is better to advise avoiding significant caffeine doses as they can reduce the drug’s effectiveness via mechanisms that are not yet uncovered [36].

### 2.3. Prevalence of Lamotrigine Use in Medical Treatment

#### 2.3.1. Therapeutic Application of Lamotrigine in the Management of Epilepsy

Lamotrigine, a second-generation antiepileptic drug, is a pharmaceutical agent that is widely used to treat various types of seizures [16]. Lamotrigine is utilized in both monotherapy and adjunctive treatment settings. In its monotherapy formulation, it is employed to address partial (focal) seizures, encompassing both simple and complex manifestations, as well as generalized seizures [37]. Generalized tonic–clonic seizures, whether primary or secondary, are indicative of lamotrigine treatment for adult patients and children who have reached the age of 12 [38]. Complementary treatment with lamotrigine is employed in children who have reached the age of 2 and are afflicted with partial or generalized seizures [39]. The pharmaceutical agent is also utilized as an adjunct therapy in a severe, drug-resistant form of epilepsy in patients diagnosed with Lennox–Gastaut syndrome [40]. The primary objective of the therapeutic intervention is to avert the occurrence of seizures. That is to say, the intervention is long-term, with the objective being to curtail seizure frequency and severity [41].

#### 2.3.2. Therapeutic Application of Lamotrigine in the Management of Bipolar Affective Disorder

The observed mood improvement in patients treated with lamotrigine for epilepsy provided the rationale for evaluating the efficacy of this drug as a maintenance therapy aimed at preventing recurrent depressive episodes in patients with bipolar disorder (BD) during euthymia [1]. Despite the lack of formal approval, lamotrigine is utilized in the management of rapid-cycling bipolar depression, particularly in Bipolar II Disorder, as well as in maintenance therapy for Bipolar I Disorder [42]. Due to its limited antimanic efficacy, lamotrigine is particularly indicated for patients with a predominance of depressive episodes. Long-term use of lamotrigine in patients with bipolar disorder (BD) prolongs the time to depressive relapse, highlighting its effectiveness as a maintenance treatment [43]. Adequate dose adjustment is crucial to avoid adverse effects that could otherwise lead to treatment discontinuation [2]. Pregnancy and the postpartum period, characterized by an increased risk of mood disturbances and recurrence of bipolar disorder symptoms, do not contraindicate the use of lamotrigine [44]. The results of the meta-analysis conducted by Pariente et al. [45] confirm the favorable safety profile of lamotrigine monotherapy during pregnancy. The meta-analysis included 21 studies evaluating the effects of LTG on pregnancy outcomes, with 1412 patients diagnosed with bipolar disorder and a healthy control group of 774,571 women. The analysis did not indicate an increased risk of congenital malformations associated with LTG monotherapy. No significant increase in the risk of miscarriage, stillbirth, preterm birth, or small-for-gestational-age (SGA) neonates was observed compared to the general population [45]. Compared with other mood stabilizers, lamotrigine monotherapy is associated with a lower risk of congenital malformations and does not affect the neurological development of children exposed to lamotrigine in utero [44].

#### 2.3.3. Therapeutic Application of Lamotrigine in the Management of Personality Disorders

Lamotrigine use also appears to offer potential benefits in BPD. Lamotrigine therapy helps patients cope with mood instability, and mood stabilization in turn augments therapeutic outcomes, including reductions in impulsivity and emotional dysregulation [46]. The efficacy of symptom alleviation in BPD among individuals treated with lamotrigine creates more favorable conditions for both patients and therapists during dialectical behavior therapy (DBT), resulting in improved therapeutic outcomes [47]. The efficacy of lamotrigine in reducing core symptoms of BPD has thus far been confirmed primarily in short-term, small-scale studies. Reich et al. conducted a 12-week, double-blind study involving 15 patients receiving lamotrigine and 13 patients in the placebo-controlled group, demonstrating a significant reduction in impulsivity, as measured by the ZAN-BPD scale, in the lamotrigine-treated group [48]. In a subsequent study, Tritt et al. [49] assessed the impact of lamotrigine on aggression reduction in female patients with BPD. The study was randomized, double-blind, and placebo-controlled, lasting eight weeks, during which 18 patients received lamotrigine, while 9 received a placebo. The results indicated a statistically significant decrease in aggression levels among patients treated with lamotrigine for eight weeks alongside improved emotional regulation compared to the placebo group [49]. Given the promising therapeutic potential of lamotrigine in BPD treatment, further in-depth and long-term studies with broader inclusion criteria are required to comprehensively elucidate its effects and potential clinical benefits in this population.

#### 2.3.4. Off-Label Treatment with Lamotrigine

There are reports in the literature indicating that lamotrigine use has positively influenced prophylactic treatment of basilar migraine with aura, panic disorder, binge eating disorder, and restless leg syndrome [50]. In cases of trigeminal neuralgia that is resistant to first-line therapies, or when patients are intolerant to carbamazepine or oxcarbazepine, lamotrigine may be employed [51]. Furthermore, additional off-label applications include the treatment of peripheral neuropathy, neuropathic pain, and fibromyalgia [52].

## 3. Principles of Pharmacotherapy in Bipolar Disorder (BD)

Lamotrigine is among the most thoroughly researched medications used for the treatment of depression in patients with bipolar disorder. It has been shown to be both safe and well tolerated by patients compared to placebo [53]. At the same time, it does not increase the risk of manic episodes [43]. The main benefits of lamotrigine in BD are prevention of depressive episodes and prevention of rapid phase changes (rapid cycling) [54]. Lamotrigine has been demonstrated to prolong periods of remission and reduce the intensity and frequency of recurrent depressive episodes. Its prophylactic and stabilizing effects make it a key drug in the long-term treatment of BP. Lamotrigine is used in both monotherapy and combination therapy in the treatment of BP types I and II [55,56].

### 3.1. Lamotrigine Monotherapy

Lamotrigine monotherapy is recommended in the pharmacological treatment of patients experiencing a depressive episode in the course of BD Type I or II, particularly those with predominantly depressive phases. It is also indicated for individuals with predominantly manic/hypomanic episodes or phase changes following prior antidepressant treatment. Furthermore, lamotrigine monotherapy may be utilized in patients presenting with a mixed depressive state or in rapid-cycling BD Type II. This monotherapy approach is especially applied in maintenance treatment for BD Type II. Despite its limited efficacy in preventing manic symptoms, lamotrigine is also employed in maintenance therapy for BD Type I when depression predominates [57,58].

### 3.2. Combination Therapy as a Response to Clinical Diversity

Bipolar disorder is characterized by alternating manic, depressive, and mixed episodes [59]. The complexity of symptoms and variability in the course of the disease often render monotherapy insufficient—particularly in rapid-cycling BD, in maintenance treatment for both BD Type I and II, and in mixed manic states. The limited efficacy of monotherapy under these conditions underlies the rationale for using lamotrigine in combination therapy [60,61].

### 3.3. Combination Therapy with Lithium

In the maintenance treatment of BD Type I, lamotrigine can be administered in conjunction with lithium. Lithium, classified among the classic mood stabilizers, is effective in both treating mania and preventing depressive episodes. Its antidepressant action is further supported by lamotrigine, thereby enhancing the prevention of depression and contributing to mood stabilization [62]. In Bipolar Disorder Type I with predominant depression, lamotrigine can be combined with quetiapine—an atypical antipsychotic effective in preventing both manic and depressive episodes during maintenance therapy. A similar approach may be employed in the maintenance treatment of BD Type II with rapid cycling, where lamotrigine is combined with a classic mood stabilizer or an atypical antipsychotic. Combination therapy is also indicated as a second-line option for treating mixed manic states (combining lamotrigine with olanzapine) and as a third-line approach (combining clozapine with lamotrigine) [57].

### 3.4. Combination Therapy with Valproic Acid

Another combination therapy option involves the use of lamotrigine together with valproic acid. Among patients who received dual therapy, a marked improvement was noted in preventing depressive relapses in Bipolar Disorder (BD) Type I. In BD Type II patients with rapid cycling, this approach effectively stabilized mood, reduced relapse frequency, and prolonged remission periods [61]. Patients who exhibit intolerance to lithium or valproic acid may be eligible for a combination of lamotrigine and carbamazepine [63]. Before initiating combination therapy with valproic acid or carbamazepine, it is essential to verify that the patient is not pregnant, as both valproic acid and carbamazepine have teratogenic effects that significantly limit their use during pregnancy [64].

### 3.5. Combination Therapy with Antidepressants

In both BD Type I and, especially, BD Type II—which is characterized by predominantly depressive episodes—it is possible to implement a combination regimen of lamotrigine and antidepressants [65]. Particularly severe cases of bipolar depression that prove resistant to standard treatments may require combination therapy involving lamotrigine as a mood stabilizer and antidepressants with a low risk of inducing mania, such as paroxetine (Selective Serotonin Reuptake Inhibitor—SSRI) or bupropion (Norepinephrine–Dopamine Reuptake Inhibitor—NDRI). This combined treatment approach diminishes the severity of depressive symptoms and reduces their recurrence rate [66].

## 4. Dosage, Serum Concentration and Therapeutic Efficacy

### 4.1. Monotherapy or Combination Therapy with Non-Inducers/Inhibitors of Glucuronidation

For adult patients with bipolar disorder receiving lamotrigine monotherapy, the recommended initial dose is 25 mg per day during the first two weeks to achieve the full maintenance daily dose. In the third and fourth weeks, the dosage is increased to 50 mg per day, administered either as a single dose or divided into two doses of 25 mg each [67]. The dose increase from 25 mg to 50 mg is intended to prepare the body for the maintenance dose. In the fifth and sixth weeks, the dosage is increased to 100 mg and 200 mg, respectively. The target maintenance dose is 200 mg per day. However, depending on the individual clinical response, it may be adjusted within a range of 100–400 mg [68].

### 4.2. Combination Therapy with Valproate

For combination therapy involving lamotrigine and valproate (an inhibitor of lamotrigine glucuronidation), the initial dose should be 12.5 mg per day (administered as 25 mg every other day) for the first two weeks. During weeks three and four, the dose should be increased to 25 mg per day. Starting in the fifth week, the dose can be increased to 50 mg per day. The target maintenance dose is typically 100 mg per day, administered once daily or in two divided doses. Depending on clinical response, the dose can be increased up to a maximum of 200 mg per day [69].

### 4.3. Combination Therapy with Carbamazepine

In combination therapy with lamotrigine and carbamazepine (an inducer of lamotrigine glucuronidation), the initial dose is 50 mg per day for the first two weeks. During weeks three and four, the dose should be increased to 100 mg per day, administered in two divided doses. Starting in the fifth week, the dose should be gradually increased by up to 100 mg every one to two weeks until a maintenance dose of 200–400 mg per day is achieved, administered in two divided doses [70].

### 4.4. Therapeutic Dose Ranges and Efficacy

Kumar et al. [71] conducted an analysis on the optimal serum levels of lamotrigine and its therapeutic efficacy in patients with bipolar disorder and severe depression. The analysis included seven studies involving 226 patients with bipolar disorder and 17 patients with severe depression. Inclusion criteria required that the study duration range from 6 to 96 weeks. The therapeutic daily dose of lamotrigine among patients varied from 25 to 425 mg per day. The study results revealed inconsistencies in the relationship between serum lamotrigine levels and therapeutic efficacy. Three out of the seven analyzed studies did not find a significant correlation between lamotrigine levels and clinical improvement. However, two studies (n = 99) confirmed that serum concentrations exceeding 3.25 μg/mL were associated with better therapeutic outcomes. One study (n = 25) indicated that lamotrigine remained effective and relatively safe within a broad serum concentration range of 5 to 11 μg/mL. A cross-sectional analysis conducted by Kumar et al. [71] suggests that the minimum threshold serum concentration required for therapeutic efficacy is 3 μg/mL. However, data regarding the therapeutic range remain inconsistent, preventing the establishment of definitive clinical guidelines. The authors highlight the lack of consistent evidence to determine the optimal serum concentration range of lamotrigine for treating bipolar disorder. Further studies involving larger patient cohorts are necessary to more accurately assess the relationship between serum lamotrigine levels and its therapeutic effectiveness [71].

### 4.5. Lamotrigine Side Effects

As with every drug, adverse effects have to be accounted for, and knowledge about their possible presence is of significant therapeutic value. First and foremost it is needed to assess LTG’s risk of cutaneous reactions as they are the most prominent and well-known adverse effect of said agent because of their prevalence and diversity and, in some unfortunate cases, even lethality. SJS and TEN are the most feared, but the incidence of DRESS, MPE and idiosyncratic and nondrug-related skin reactions is as important because those reactions can lead to redundant therapy discontinuation resulting in rebound convulsions [72]. Thus, it is rational to educate healthcare workers on differentiation between these cutaneous reactions.

A considerable number of studies showed that placebo had a comparable probability of inducing urticaria compared to LTG, which clearly indicates that rash in a significant amount of cases could be caused just by injection on its own [72]. On the positive side, the occurrence of more dangerous reactions tapered off following 1994, which was later confirmed to be caused by reductions in the starting dose. Thus, titration rate and dosing are key for safe therapy with LTG [18]. Additional risk factors and factors that can complicate diagnosis are concomitant viral infections which can both precipitate and be misdiagnosed as hypersensitivity. However, some symptoms like lymphadenopathy, fever and mucous membrane involvement should be taken as seriously as possible since they can prelude systemic disturbances seen in DRESS, SJS and TEN. Such symptoms are regarded as definitive contraindications to further therapy, at least for the time of their presence. Typical flu-like symptoms often precede blister and cutaneous lesion formation. If the complications are not severe, reintroduction (with altered dose and titration) could be beneficial in the majority of cases, especially since these reactions are postulated to be characterized by desensitization mechanism [72]. However, it is important to note that the study which concluded this took into the account symptoms even as non-significant as single oral lesions in addition to MPE and others [72]. Hemophagocytic lymphohistiocytosis (HLH), another rare, immune-related drug reaction that occurs especially in children, has been observed in few cases of LTG therapy, but it is by no means as important in LTG therapy as SJS [73].

The molecular mechanism of rash induction is needed to be studied more extensively. Nevertheless, sulfatases (the only biotransformation enzymes abundant in skin tissue) are being evaluated as possible culprits for the formation of an inflammation-inducing metabolite. As sulfates are a good leaving group, sulfated LTG metabolites could readily react with proteins, consequently forming antigens responsible for immune/inflammatory reaction [74]. Apart from dermatological complications, other and often non-specific adverse effects occur, particularly CNS malfunctions such as dizziness and somnolence. Such effects rarely persist and can lead to seizures or behavioral changes, but those cases constitute the minority. Overall, type B adverse effects (idiosyncratic ones) were particularly more commonly seen than type A effects (related to the drug’s known mechanism of action) [18,75]. HCN channels that may be crucial to LTG’s therapeutic profile are also a cause of its controversies regarding cardiac morbidity [18]. Even though it is advised to inform patients about such effects, they are much less pronounced than during therapy with alternative agents such as valproate or carbamazepine [14,18,76]. Lamotrigine has not been found to affect weight, which is especially important in BD patient handling, especially since other euthymic agents like valproate are known to predispose individuals to developing metabolic syndrome with concomitant weight gain. The destructive behavior of patients with bipolar disorder puts them at increased risk of developing DM, which means that therapeutic agents that have a negative impact on weight should not ideally be taken into account when choosing the right therapy [77].

At present, LTG is not considered teratogenic and is not associated with overall neurodevelopmental disturbances [78], but the number of studies on this topic should ideally be increased as LTG has gained a lot of popularity in therapy in recent times. The aspect of teratogenicity is of specific interest within the context of BD patients, as they are at increased risk of unplanned pregnancy [18,78]. Regarding the aspect of breastfeeding, the drug can invade infant circulation and its exposure can be as high as 50 percent of that of the mother/parental one. However, no significant complications of this interactions were reported and monitoring such infants is, for now, the only advised action to take [78]. There are a few actions one could take to bring a patient out of a state of intoxication with LTG [29]. As this drug’s effectiveness depends on its plasma level, a logical approach could involve sequestration of the drug in lipid droplets or performing plasma dialysis. A case study by Li et al. [29] suggests an advantage of continuous renal replacement therapy (CRRT) over intermittent hemodialysis because the latter can precipitate drug concentration spikes, which could lead to rebound toxicity. Another approach makes use of LTG kinetics: by administering rifampin, a liver enzyme inducer, LTG metabolism is increased and, subsequently, its clearance goes up. Blood alkalization with sodium bicarbonate has also been used in the past. Nevertheless, said studies suggest the superiority of lipid emulsions, dialysis and rifampin over bicarbonate injection [29].

### 4.6. Incidence of SJS as an Adverse Effect of LTG Administration

A retrospective study comprising 83 cases of severe cutaneous adverse reactions (which included SJS, TEN, and their overlap) carried out by About-Taleb et al. showed oral LTG therapy to be a culprit in 6% (6 cases), which falls behind the result for carbamazepine (14.5%) and valproate (8.4%) oral formulations [79]. A study by Abtahi-Naeini et al. on an Iranian population (n = 100) showed lamotrigine to be one of the most common causative agents for SJS in the adult group; however, for the pediatric population, there was a greater association with antibiotic and phenobarbital usage [80]. Huang et al. presented in their research the incidence of LTG-induced SJS/TEN as n = 30 (3%), in comparison with carbamazepine’s n = 101 (11%) [81]. Glahn et al. pointed out in their study that increasing off-label lamotrigine usage correlated with a higher incidence of SJS/TEN; nevertheless, antibiotics were also the most frequent causative agents. It should be kept in mind, however, that only 30 cases were analyzed in said study [82]. On the other hand, research by Moshayedi showed antiepileptics to be more harmful in terms of SJS incidence compared to antibacterial agents, with LTG contributing to 17% of these reactions. Such a result was achieved and surpassed only by carbamazepine [83]. A significantly more large-scale analysis was performed in 2024 by Bataille et al., making of use of the WHO database (VigiBase), although it included only a pediatric population (under 18 years old). Overall, 7342 patients diagnosed with SJS/TEN were included and 165 pharmaceutical agents were shown to be involved in its pathogenesis, with LTG again achieving second place in terms of being the cause (n  =  780, 10.6%), again falling behind carbamazepine (n  =  858, 11.7%) [84]. The ALDEN algorithm, the main topic of a study by Sassolas et al., which was developed with greater reliability in mind, when compared with case–control analyses regarding the drug–SJS relationship, showed LTG to have a definitive association with the disease, with the correlation being very strong [85]. This result even further sets in stone the topic of our paper. Regarding of risk estimation during LTG therapy, as Fukasawa et al. showed, out of 4743 Japanese patients, 4 people developed SJS and/or TEN, which roughly translates to 84.33 cases per 100,000 users [86]. Another analysis concluded the risk to be 2.82/10,000 new users, using the computerized database of Clalit Health Services as the data source [87].

## 5. Stevens–Johnson Syndrome Characteristics

The rare and potentially fatal skin conditions known as Stevens–Johnson Syndrome (SJS) and toxic epidermal necrolysis (TEN) have an incidence rate of 0.5–1.4 million per year and a mortality rate of 25–35% [88]. They predominantly occur as a side effect after administration of pharmaceuticals from groups such as antibiotics, nonsteroidal anti-inflammatory drugs (NSAIDs) and antiepileptic drugs—including lamotrigine [89,90]. Both conditions share common pathogenesis and their differentiation is chiefly based on the percentage of body surface area (BSA) affected. While TEN causes epidermal detachment of over 30% of the BSA, SJS affects less than 10% of total skin surface. TEN/SJS overlap is a condition diagnosed when the BSA affected lies in the range between SJS and TEN [11].

A mnemonic can aid in the early diagnosis of SJS/TEN. Each letter in the acronym “SJS/TEN” corresponds to a specific diagnostic criterion. The first “S” denotes systemic symptoms including fever exceeding 39 °C, general malaise and other related manifestations. “J” represents jarring pain, characterized by elevated skin tenderness observed in SJS. The second “S” refers to the involvement of mucosal surfaces, such as the oral, ocular and genital mucosa, which are commonly affected. “T” stands for atypical target lesions describing an unusual appearance of lesions typical for SJS, as described previously. “E” signifies Ear–Nose–Throat (ENT) symptoms, including odynophagia, dysphagia, nasal obstruction or other manifestations localized near the oral cavity. The final letter, “N”, encompasses two criteria: the resent recent addition of a new drug to the patient’s pharmacotherapy, like lamotrigine, and the presence of a positive Nikolsy’s sign [91]. This mnemonic proves to be an effective tool used in the identification of SJS and TEN, enabling early intervention and potentially enhancing patient prognosis.

### 5.1. Pathogenesis

SJS/TEN is mainly induced by a type IVb hypersensitivity reaction mediated by cytotoxic T lymphocytes (CTLs) and natural killer (NK) cells (Figure 2). This T cell-dependent immune response requires both an antigen-specific signal via the T cell receptor (TCR) that requires HLA-bound immunogenic peptides and a co-stimulating signal involving CD28 on T cells and B7 (CD80/CD86) on antigen-presenting cells (APCs) [92].

The signaling cascade leads to T cell activation and proliferation and triggers the release of cytotoxic mediators and inflammatory cytokines, mainly IL-4 and IL-5, which promote eosinophil proliferation, activation, and chemotaxis, driving cell apoptosis [93]. After resolution, the formation of memory T cells occurs, allowing future responses to the same antigen or HLA complex upon re-exposure, without the necessity of co-stimulation, yielding rapid and effective response. Nevertheless, inappropriate T cell response to harmless stimuli, including numerous medications, can lead to autoimmunity, drug hypersensitivity reactions (DHRs) and delayed-type drug hypersensitivity reactions (DTHRs) [92,93].

Other hypersensitivity subtypes to blame for the pathogenesis of these reactions include type IVc. Driven by drug-specific cytotoxic CD8+ T cells, apoptosis occurs by granulysin, perforin, granzyme B or Fas/FasL signaling. Granulysin is a cytotoxic protein found in CTL and NK granules that was proven to induce keratinocyte apoptosis via ER stress and caspase-7 activation in SJS/TEN. On the other hand, granzyme B is capable of both caspase-independent and caspase-dependent apoptosis via procaspase-3 activation, cleavage of Bid, cytochrome release or DNA fragmentation [93]. Additionally, activation of Fas receptors, present on keratinocyte surfaces, by ligands presented by lymphocytes; further accelerates apoptosis by initiation of a signaling cascade. It happens by attracting the adaptor protein FADD (Fas-associated death domain). This leads to the assembly of a multiprotein platform known as the death-inducing signaling complex (DISC). Within this complex, caspase-8 is recruited and undergoes structural changes through self-activation, which enables it to initiate a cascade involving other caspases. These downstream effector caspases execute the final molecular events that culminate in programmed cell death [94,95]. Possible mechanisms of keratinocyte death pathogenesis are illustrated in Figure 3. The progression of keratinocyte death results in widespread skin damage, characterized by epidermal necrosis and detachment [96].

Tissue specificity in T cell-mediated hypersensitivity reactions remains an under-explored field. One of the most striking features of these reactions is their predilection for the skin, which serves as the most frequently affected organ, despite systemic drug exposure. This phenomenon probably arises from interactions between various factors including local drug metabolism, the skin’s unique immune environment, tissue-specific antigen presentation or immune memory. Skin presents a special immune environment characterized by constitutive danger signaling and resident immune cells, for example, dendritic or Langerhans cells, favoring immune activation over tolerance [13,97]. Another possible mechanism can result from higher levels of sulfotransferase (SULT) expression in skin; compared to other metabolic pathways. LTG N-sulfate, the metabolite formed by SULT activity, is chemically reactive and can form covalent bonds with alcohol and carboxylic acid groups, including those on amino acids such as serine and tyrosine. Elevated levels of serine amino acids within the keratin-dense layers of the epidermis allow many potential sites for binding. Unlike other reactive intermediates, LTG N-sulfate does not undergo detoxification by glutathione, which increases its potential for protein binding. Observations of tyrosine sulfation further indicate that LTG N-sulfate may induce protein modifications that contribute to immunogenicity and hypersensitivity [14,74].

### 5.2. Clinical Features

SJS frequently begins with a prodromal phase characterized by fever and generalized malaise, which persists for several days. This is followed by the rapid onset of a symmetric erythematous maculopapular rush, often associated with mucosal involvement. The initial cutaneous lesions present as ill-defined erythematous macules with central purpura, which may coalesce as the condition progresses [11]. The lesions commonly exhibit a positive Nikolsky sign, which is characterized by the formation of blisters and the separation of the epidermis from the underlying dermis upon the application of light pressure [11,98]. Due to its distinctive histological characteristics, biopsy is a useful technique for diagnosing SJS. Samples presenting an epidermis on tissue sections are evaluated. Epidermal pathologies including its thickness, necrolysis stage or composition of present dermal infiltrates are assessed [99]. Full-thickness necrosis, orthokeratosis and development of subepidermal cleft covering the dermis align with diagnostic features of SJS [100].

### 5.3. SJS Differential Diagnosis

Although SJS is characterized by its distinct pathophysiology, its differentiation from other drug-induced dermal reactions necessitates the use of precise diagnostic tools. This is especially relevant considering the existence of several medical conditions presenting similar clinical manifestations, despite having distinct underlying mechanisms and features. Differential diagnosis of SJS is particularly crucial in cases of erythema multiforme (EM), Staphylococcal Scalded Skin Syndrome (SSSS), linear IgA bullous dermatosis (LABD), graft-versus-host disease (GvHD), and other rashes with seemingly similar clinical features [101,102,103,104]. The initial and most crucial step in the diagnosis of SJS involves exclusion of these conditions. This is typically achieved through an extensive evaluation of the patient’s clinical history, meticulous review of any recent changes in pharmacotherapy and the performance of targeted diagnostic tests [101].

The etiology serves as a key differentiating factor among the conditions discussed. As mentioned earlier, SJS and TEN are caused by exposure to certain groups of drugs [11,89,90]. EM is typically triggered by infections, particularly Herpes Simplex Virus (HSV) [105]. In contrast, SSSS is caused by toxins produced by Staphylococcus Aureus [106]. LABD is an autoimmune disorder characterized by the deposition of IgA antibodies at the basal membrane of the dermoepidermal junction, leading to the formation of subepidermal blisters [107]. Biopsy findings reveal additional distinguishing features among these conditions. As previously mentioned, SJS histologically demonstrates full-thickness necrosis, orthokeratosis; and the formation of a subepidermal cleft that separates the epidermis from the dermis, which are characteristic diagnostic features of SJS [100]. In SSSS, epidermal cleavage occurs at or just below the stratum granulosum, a feature that can resemble the histopathological findings seen in Pemphigus foliaceus [108]. The only cells identifiable are acantholytic epithelial cells (acanthocytes) from the granular layer and the lamellar structures of the stratum corneum, with no evidence of microorganisms, inflammatory cells, or apoptotic bodies in the affected tissue [109]. In LABD, biopsy reveals the formation of subepidermal blisters containing fibrin and an infiltrate of neutrophils. The upper dermis shows a mixed inflammatory infiltrate, primarily composed of neutrophils and lymphocytes, surrounding blood vessels and extending throughout the interstitial space, with a notable predominance of neutrophils in the superficial dermis [108,110]. Finally, in EM, histopathological observation reveals lymphocytic exocytosis, necrotic keratinocytes, and basal cell vacuolar degeneration, which are key features of the condition [101].

Recent research has revealed promising biomarkers for SJS/TEN, offering new possibilities for early diagnosis and more precise treatment [90,111]. One of the studies analyzed peripheral blood mononuclear cells (PBMCs) from individuals that recovered from SJS/TEN by re-exposing them to the causative drug. Through proteomic analysis, galectin-7 was identified as a potential biomarker, with notably higher levels found in the serum of SJS/TEN patients compared to those suffering from non-severe Cutaneous Adverse Drug Reactions (cARDs). The levels of galectin were found to correlate with disease severity, being elevated in the acute phase and diminishing after the reaction subsided [90,111]. Receptor-interacting kinase-3 (RIP3), a key mediator of necroptosis, has been identified as another possible biomarker for SJS/TEN. Higher serum levels of RIP3 were detected in SJS/TEN patients, especially during the acute phase, and were linked to the severity of the disease [90,93,112].

### 5.4. Lamotrigine-Induced SJS/TEN and Genetics

While the onset of SJS is primarily precipitated by specific pharmacological agents, recent research has confirmed that the presence of specific alleles of the HLA gene notably amplifies patients’ susceptibility to developing the condition. These genes, located on chromosome 6, are characterized by considerable allelic polymorphism, which contributes to the variability in immune responses across different populations [113]. HLA alleles are classified into Class I and Class II categories, each of which is specialized in presenting antigenic peptides to T cells, thereby initiating the activation of the immune response [112]. Certain HLA alleles (HLA-B*15:02 and HLA-A*31:01) are associated with elevated risk of developing Cutaneous Adverse Drug Reactions (cARDs), with SJS and TEN being notable examples [113,114]. The strongest association with drug-induced SJS/TEN was found with the HLA-B*15:02 allele, and it was most pronounced in Asian populations, with a substantial body of research confirming the link, while only limited evidence from Caucasian populations suggests a similar risk [114,115]. Furthermore, the influence of specific HLA alleles on the development of Cutaneous Adverse Drug Reactions (cADRs) varies depending on the drug in question, with the level of association differing across different pharmaceuticals [101,112,113,115]. HLA-B 58:01 is a critical genetic marker for allopurinol-induced SJS/TEN, particular particularly in populations of Han Chinese or Japanese decent. In contrast, the HLA-B*15:02 allele is found at much lower frequencies in European and Hispanic populations, with a weaker association with allopurinol-induced SJS/TEN in these groups [101]. In Thailand, the research revealed a significant genetic association between severe ocular development and numerous HLA types. These predominantly included HLA-A33:03, HLA-B44:03, HLA-C07:01 and the HLA-B44:03–HLA-C*07:01 haplotype [116]. In Brazil, on the other hand, the condition was mostly associated with HLA-B44:03 and HLA-DQB104:02 [116]. Over a decade ago, the FDA recommended HLA-B*15:02 genotyping prior to carbamazepine use, leading to a notable reduction in the occurrence of CBZ-induced SJS/TEN [93]. This highlights the significant ethnic variability in genetic susceptibility to drug-induced severe cADRs. A study by Koomdee et al. [117] found that HLA-A02:07 was present in 33.3% of LTG-induced CADR patients and showed significantly higher frequencies than both the treatment control and general population groups, although the sample was undeniably small (n = 15), and the authors emphasized the need for further research before a new screening marker in the form of 02:07 allele becomes recognized as prognostic enough. Furthermore, in the subgroup analysis specifically for LTG-induced SCAR (which includes SJS), the association between HLA-A02:07 and SJS did not reach statistical significance, with the small sample being even more pronounced here [117]. As mentioned in the pharmacokinetics section (Section 2.2), UGT enzyme polymorphisms may be crucial for patient-to-patient differences in clinical response and toxicity and, thus, SJS development. Notably, UGT1A4 and UGT2B7 isoforms are said to be involved, and drugs that interfere with UGT enzyme efficacy are able to modulate LTG’s AUC; (that is, exposure and subsequent risk) for SJS development [27,86]. Typical agents being brought up are valproic acid, carbamazepine, and phenytoin. Flunitrazepam and clonazepam were concomitantly administered and such an interaction was highly probable to take place, but nevertheless, a sample consisting of two patients should not lead to generalized conclusions [117].

While not a major risk marker, HLA-B38:01 has been linked in some cases to increased susceptibility to SJS/TEN when triggered by certain medications. An association exists between the HLA-B38:01 allele and an elevated likelihood of SJS/TEN triggered by certain antiepileptic drugs in the Spanish population. Carriers of that specific allele were more susceptible to developing SJS/TEN, with odds ratios ranging from approximately 13 to 147 depending on the control group [118]. By contrast, a European study revealed a limited correlation between the HLA-B*38 allele and SJS/TEN development, observed in a small number of cases involving lamotrigine (19 cases) or sulfamethoxazole (28 cases). This suggests lower clinical significance of the HLA-B38:01 allele; compared to the previously mentioned HLA-B58:01, where, in the same study, 61% of affected individuals carried the allele and the odds ratio reached 80 [119]. Statistical correlation was identified between the HLA-A*24:02 allele and lamotrigine-induced SCAR in the southern Han population. Individuals who possessed this allele exhibited a more than fourfold increased likelihood of developing LTG-induced SJS/TEN (45.5% in those with the HLA-A*24:02 allele vs. 15.7% in those without the allele). HLA-A*24:02 has been identified as a significant risk factor for SJS/TEN in patients treated with the most commonly used aromatic ring antiepileptic drugs (lamotrigine, carbamazepine, phenytoin). The data presented by Yi-Wu Shi et al. served as the basis for the implementation of pretreatment screening in southern China [120]. A similar study was conducted in the Korean population. Koreans carrying HLA-A*24:02 were associated with an approximately 2–3-fold but statistically insignificant increase in the risk of LTG-induced SCAR (OR = 2.57; 95% CI 0.77–8.61; *p* = 0.14). This result highlights the need for studies on a larger cohort of Korean patients to unambiguously assess the association of the HLA-A*24:02 allele with cADR [121].

### 5.5. SCORETEN Scale

The Severity-of-Illness Score for Toxic Epidermal Necrolysis (SCORETEN) is a commonly utilized method for evaluating the potential of death in individuals suffering from severe skin conditions like SJS or TEN. It serves to calculate the risk by evaluating seven distinct factors including the patient’s age, degree of skin damage, history and presence of malignancy and other medical parameters (Table 1) [112,122]. While it is clear that the scale serves as an efficient tool for diagnosing SJS/TEN patients, new research proposes its limitations in accurately forecasting mortality [123,124]. The findings of a decade-long retrospective study performed in Wuhan suggest that the ABCD-10 score (age, bicarbonate, cancer, dialysis, 10% body surface area) demonstrates superior predictive accuracy in the context of Chinese healthcare settings, outperforming the SCORETEN scale [124]. Another study conducted in India suggests numerous modifications to the SCORETEN scale, including addition of risk factors such as elevated blood urea nitrogen, thrombocytopenia, and elevated liver enzymes [125].

Additionally, research advocates for adapting SCORETEN to account for regional differences by considering local factors; like pneumonitis, which was found to be linked to fatal outcomes in this particular study [125]. However, most recent data disprove the ABCD-10 score’s supremacy in mortality assessment over SCORETEN [126]. Finally, Re- SCORETEN scale was proposed, in which, in addition to seven initial factors, red cell distribution width to hemoglobin ratio (RDW/Hb) is evaluated to assess the mortality in SJS/TEN patients [127].

### 5.6. SJS Pharmacotherapy

Owing to its drug-induced character, the management of SJS/TEN shall begin with the immediate discontinuation of any suspected drugs, including lamotrigine, to halt progression. Pharmacotherapy focuses on supportive care, including fluid and electrolyte resuscitation, preventing infection and management of pain. Additionally, immunomodulatory treatments may be considered, depending on the patient’s condition [128]. Traditionally, apart from discontinuation of causative drug administration, treatment involved introduction of systemic glucocorticoids and intravenous immunoglobulin (IVIG). Nevertheless, it was recently shown that calcineurin inhibitors such as cyclosporine have promising potential in SJS/TEN therapy due to its anti-apoptotic properties. Its mechanism of action is based on inhibiting T-cell activation and reducing the production of caspases and cytokines involved in apoptosis like TNF-α leading to decreased keratinocyte apoptosis. Clinically, early administration of cyclosporine has been linked to decreased speed of disease progression and quicker re-epithelialization [88,129]. Other TNF-α inhibitors used for SJS/TEN management include infliximab or etanercept [130]. Immune checkpoint inhibitors used in cancer treatment, like pembrolizumab, have been linked to the occurrence of SJS/TEN, suggesting the importance of early diagnosis and immediate therapy including cyclosporine. The immunological characteristics of SJS/TEN pathophysiology have prompted the research community to focus pharmacotherapy on targeting the underlying autoimmune pathways driving disease. Recent studies have highlighted the potential of Janus Kinase inhibitors (JAKi) as an effective tool in the management of SJS/TEN. Researchers have identified the JAK/STAT and interferon signaling pathways as a key drivers of the disease. Using proteomic analysis, activation of these pathways in keratinocytes and immune cells of SJS/TEN patients was measured. This showed significant JAK/STAT pathway activation in both cell types, with upregulation of proteins such as STAT1 playing a key role in interferon signaling. Clinical trials revealed that patients treated off-label with JAKi, including abrocitinib, experienced halted disease progression and demonstrated re-epithelialization of the affected areas [131]. In clinical settings, excessive antibacterial prophylaxis is crucial, since pathogens such as Staphylococcus aureus and Pseudomonas spp. frequently colonize wound sites in affected patients. To minimize the risk of developing antibiotic resistance in hospitalized patients, aseptic techniques are favored over prophylactic antibiotics that should be reserved only for severe cases [132]. This approach is particularly crucial given the fact that some sulfonamide antibiotics are considered high-risk medications for SJS/TEN pathogenesis [133]. For acute pain relief, combination of opioids and nonsteroidal anti-inflammatory drugs (NSAIDs) can be employed. Long-acting opioids such as methadone along with gabapentin for neoplastic pain can be utilized for treatment of chronic and subacute pain [133].

### 5.7. Pharmacovigilance Data on Lamotrigine and Stevens–Johnson Syndrome

Pharmacovigilance (PV) is a strategic approach to monitoring drug safety and identifying adverse drug reactions (ADRs) [134]. Data from the FDA Adverse Event Reporting System (FAERS), EudraVigilance (EVDAS), and the WHO global pharmacovigilance database (VigiBase) allow for the evaluation of the frequency of adverse reactions and the detection of a potential causal link between the use of lamotrigine and the occurrence of rare but severe skin reactions, including Stevens–Johnson Syndrome [135,136,137].

According to the European Medicines Agency’s assessment report for Lamictal (lamotrigine), Stevens–Johnson Syndrome (SJS) is identified as a rare but potentially life-threatening adverse reaction. The report emphasizes that the use of excessively high initial doses or a too-rapid dose escalation, especially when combined with valproate, which inhibits the metabolism of lamotrigine, significantly increases the risk of severe skin reactions such as Stevens–Johnson Syndrome (SJS) and toxic epidermal necrolysis (TEN). The Summary of Product Characteristics (SmPC) highlights that most adverse reactions occur within the first 8 weeks of treatment; therefore, careful dose titration and close patient monitoring are essential to minimize the risk of SJS during lamotrigine therapy [138].

An analysis of data from the European EudraVigilance database dated 1 June 2025, revealed 2688 reported cases of Stevens–Johnson Syndrome associated with the use of lamotrigine in patients. Among all reported cases, 59 resulted in death, and in 547 cases, the reaction was still ongoing at the time of reporting. The highest number of cases was recorded in the 18–64 age group, with a higher incidence in women than in men. The vast majority of reports (94%) came from healthcare professionals (n = 2517), which demonstrates the high reliability of the collected data [139].

In a retrospective analysis based on data from the FAERS adverse event reporting system covering the years 1969–2024, a total of 39,398 cases related to SJS/TEN were recorded, of which 97.79% were classified as serious; and 20.86% resulted in death. The most frequently reported drug associated with SJS/TEN was lamotrigine (9.17% of cases). Most reports concerned the 18–64 age group, with women being slightly more affected than men. The high credibility of the data is confirmed by the fact that 70.38% of reports were submitted by healthcare professionals [136].

An analysis of pediatric cases (patients under 18 years of age) reported in the WHO pharmacovigilance database (VigiBase) discusses cases of toxic epidermal necrolysis (TEN) and Stevens–Johnson Syndrome (SJS) following the use of lamotrigine. The analysis includes cases reported between 1967 and 2022. Lamotrigine was one of the most frequently reported drugs associated with SJS/TEN in the pediatric population. Out of 7342 pediatric SJS/TEN cases registered in the WHO global pharmacovigilance database, 780 were linked to lamotrigine. Most reports (73%) were submitted by healthcare professionals, indicating the credibility of the collected data [84].

Despite limitations such as underreporting and the lack of data on the number of patients using the drug, pharmacovigilance activities remain essential for patient safety and the identification of adverse drug events [140]. Pharmacovigilance data indicate a strong association between lamotrigine and Stevens–Johnson Syndrome. The accumulated data underscore the importance of patient monitoring and healthcare professional education, particularly during the first eight weeks of therapy, when the risk of severe skin reactions is the highest.

### 5.8. Supportive Management of SJS/TEN

While pharmacotherapy remains the foundation in SJS/TEN management, supportive care is essential in obtaining a comprehensive treatment and a positive outcome. Antiseptic baths and topical treatments, like silver sulfadiazine, are used sparingly to prevent resistance. Specific treatment is needed for mucosal erosions affecting the mouth, genital area and eyes. It involves using antiseptic mouth washes and applying petroleum jelly to lesions in the genital area. Skin lesions should be carefully treated with non-adherent dressings, avoiding scrubbing or debridement [133]. Regular assessments by ophthalmologists should be conducted, aiming to prevent complications with the use of preservative-free moisturizers. Chronic issues including pigmentation alterations or scarring necessitate follow-up with dermatologists or plastic surgeons. Consistent application of sunscreen is essential to prevent post-inflammatory hyperpigmentation [130].

## 6. Conclusions

Lamotrigine is undeniably an agent with an impressive range of postulated mechanisms of action, primarily involving ion channel modulation and contribution to inflammatory system interference, but not only limited to them. Said versatility reveals itself in many official; (like epilepsy and BD) but also off-label uses and research on its therapeutic efficacy in disorders such as autism is ongoing. BD deserves an honorable mention because of both; the disease occurrence and increasing clinical significance, as well as the unique way in which LTG decreases the severity of this disease. LTG explicitly improves depressive symptoms of BD, making it useful for unique, specific BD subtypes. Its pharmacokinetic characteristics are not as problematic as some of its alternatives (valproate, carbamazepine) but still; are of importance; and demand some dosing adjustments in settings of concomitant antiseizure drug and OC use. Lamotrigine is safe for fetuses; nevertheless, it exhibits a variety of adverse effects, some being milder, like somnolence and dizziness, and some being of clinical emergency, such as SJS and TEN. The last two require special attention as correct decisions made by a clinician after their recognition could be life-saving, with rifampin administration and CRRT requiring a special mention in the context of the ability to lower LTG concentrations. Pharmacological intervention with TNFα and JAK inhibitors, able to minimize the destructive inflammatory reaction during SJS, is also important. Titration rate and genetic predisposition can sometimes be crucial for these reactions to occur. A clinician should be able to distinguish these conditions from many other, less fulminant skin reactions also being a possible consequence of LTG treatment. In addition, many other, non-LTG-related cutaneous reactions like SSSS, EM and LABD could complicate the diagnosis.

## Figures and Tables

**Figure 1 jcm-14-04103-f001:**
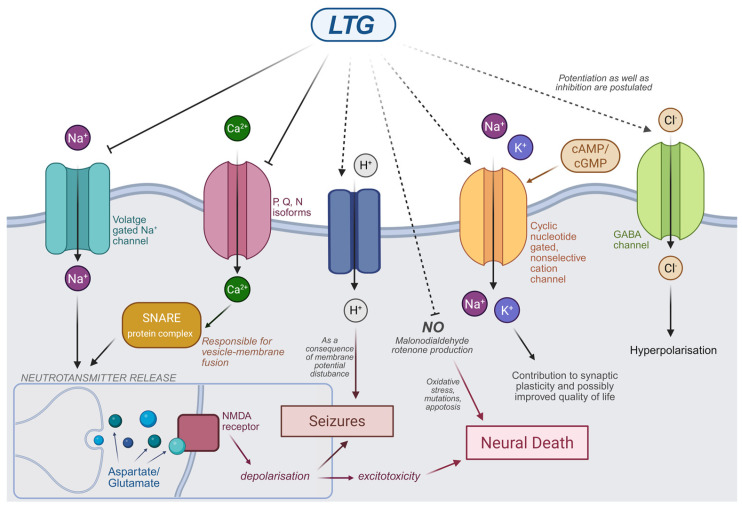
Schematic mechanism of LTG’s action (graph was prepared) (using Biorender. Kacper Żełabowski. 2025, https://biorender.com/eytrs8p (25 May 2025).

**Figure 2 jcm-14-04103-f002:**
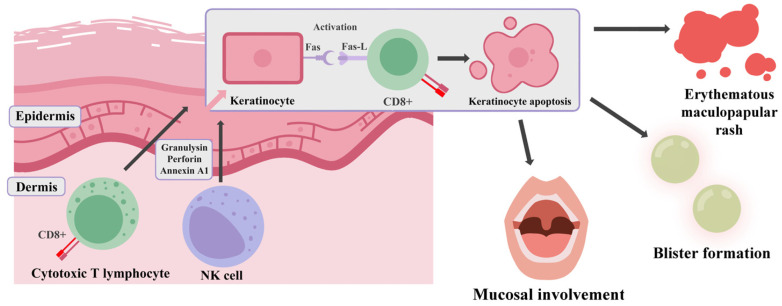
Schematic pathogenesis of Stevens–Johnson Syndrome (SJS) and toxic epidermal necrolysis (TEN). Fas—The Fas receptor (also known as CD95 or APO-1) is a type I transmembrane protein that belongs to the tumor necrosis factor (TNF) receptor family. FasL—Fas Ligand (also known as CD95L, CD178) is a type II transmembrane protein, a cytokine from the TNF family. Its interaction with the Fas transmembrane receptor (FasR, CD95, APO-1, TNFRSF6) on target cells induces apoptosis. CD8+—CD8-positive T lymphocytes. NK cell—natural killer (NK) cells (graph was prepared using Clip Studio Paint. Zuzanna Ratka.

**Figure 3 jcm-14-04103-f003:**
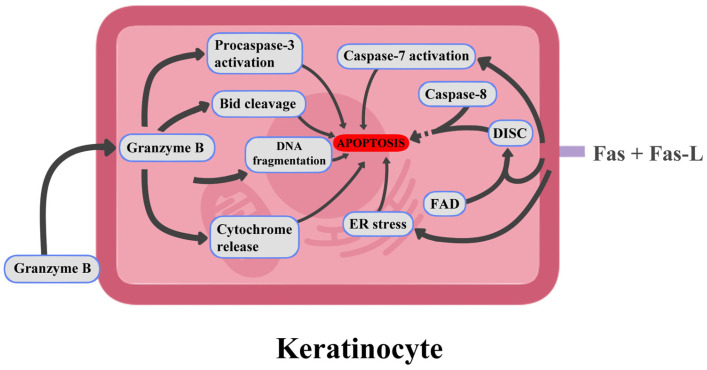
Possible mechanisms of keratinocyte death pathogenesis (graph was prepared using Clip Studio Paint. Zuzanna Ratka.

**Table 1 jcm-14-04103-t001:** Risk factors for SCORETEN and mortality rate in SCORTEN [112,122].

Risk Factors for SCORETEN	Number of Risk Factors	Mortality Rate (%)
Age above 40 years	0–1	3.2
Heart rate higher than 120 beats per minute	2	12.1
History or a present malignancy	3	35.3
Epidermal detachment area involving body surface area higher than 10%	4	58.3
Blood urea nitrogen higher than 28 mg/dL (10 mmol/L)	5	90
Blood glucose higher than 252 mg/dL (14 mmol/L)	6	90
Bicarbonate lower than 20 mEq/L	7	90

## Data Availability

Not applicable.

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
