# Peer review of "Lamotrigine Therapy: Relation Between Treatment of Bipolar Affective Disorder and Incidence of Stevens–Johnson Syndrome—A Narrative Review of the Existing Literature"

_jcm, 2025, doi:10.3390/jcm14124103_

Round 1

Reviewer 1 Report

Comments and Suggestions for Authors

The manuscript provides an overview of Lamotrigine (LTG) and Stevens-Johnson Syndrome (SJS), with three sections focused on LTG and two sections on SJS. While the authors have compiled substantial information, some of the content appears outdated, and the manuscript lacks a strong, cohesive connection between the two topics. Given the title emphasizes the relationship between LTG therapy and SJS incidence, the manuscript would benefit from a clearer focus on the direct association between LTG use and the risk of developing SJS.

  1. Sections 5.1 (SJS Pathogenesis) and 5.6 (SJS Pharmacotherapy) are too general and outdated. Please update them with current, evidence-based information. The paper [PMID: 38664435] may help.

  1. In Section 5.4 (SJS and Genetic), the manuscript mentions 4 HLA variants associated with LTG-induced SJS/TEN, but this coverage is insufficient. Numerous studies have reported genetic associations with additional genes beyond these 4 HLA alleles listed. The manuscript should be updated to include a broader range of genetic factors, both HLA and non-HLA, using information from PharmGKB and other relevant sources.  

  1. Given the manuscript's focus on Lamotrigine (LTG)-induced SJS/TEN, it would be beneficial to mention and discuss the ALDEN scoring system.

  1. The manuscript uses the term "Lyell's syndrome (TEN)," but for clarity, it would be more effective to say "Lyell's syndrome, also known as toxic epidermal necrolysis (TEN)."

Author Response

Dear Reviewer,

Thank you for the opportunity to revise our manuscript entitled "Lamotrigine therapy: Relation between treatment of bipolar affective disorder and incidence of Stevens-Johnson Syndrome - A Narrative Review of the Existing Literature" [ID: jcm-3626988].

We appreciate the time and effort the reviewer have dedicated to evaluating our work and for their constructive feedback, which has helped us improve the quality and clarity of our manuscript.

Please find below our detailed responses to each of the reviewer’s comments. We have addressed all points raised to the best of our ability and have revised the manuscript accordingly. Changes made to the manuscript are highlighted in red.

Comment: “The manuscript provides an overview of Lamotrigine (LTG) and Stevens-Johnson Syndrome (SJS), with three sections focused on LTG and two sections on SJS. While the authors have compiled substantial information, some of the content appears outdated, and the manuscript lacks a strong, cohesive connection between the two topics. Given the title emphasizes the relationship between LTG therapy and SJS incidence, the manuscript would benefit from a clearer focus on the direct association between LTG use and the risk of developing SJS”

Response:  Newly added section 4.6 recognizes the need to directly show the correlation between active LTG therapy and risk for SJS/TEN development in a patient, as well as addresses how often is LTG the cause of a SJS/TEN case in comparison with other agents with ALDEN system assessment kept in mind. Such section has potentially significant value for clinicians.

Comment: “Sections 5.1 (SJS Pathogenesis) and 5.6 (SJS Pharmacotherapy) are too general and outdated. Please update them with current, evidence-based information. The paper [PMID: 38664435] may help”

Response: The authors did not have permission to access the file, nevertheless we updated the information based on accessible resources. Section 5.1 was widely developed and considered in more detailed perspective.

Comment: “In Section 5.4 (SJS and Genetic), the manuscript mentions 4 HLA variants associated with LTG-induced SJS/TEN, but this coverage is insufficient. Numerous studies have reported genetic associations with additional genes beyond these 4 HLA alleles listed. The manuscript should be updated to include a broader range of genetic factors, both HLA and non-HLA, using information from PharmGKB and other relevant sources.”

Response: Despite extensive research, we were unable to identify additional genetic HLA associations of strong relevance. While the newly included findings from Brazilian and Thai populations do not demonstrate associations as significant as those previously reported, they contribute valuable insight by broadening the understanding of HLA involvement in the ocular complications associated with SJS/TEN. These observations underscore the potential influence of ethnic and regional genetic variability in the pathogenesis of the disease.

Comment: “Given the manuscript's focus on Lamotrigine (LTG)-induced SJS/TEN, it would be beneficial to mention and discuss the ALDEN scoring system.”

Response: Thank you for the suggestion. A description of the ALDEN scoring system has now been included in the revised manuscript to clarify its application in assessing LTG-induced SJS/TEN.

Comment: “The manuscript uses the term "Lyell's syndrome (TEN)," but for clarity, it would be more effective to say "Lyell's syndrome, also known as toxic epidermal necrolysis (TEN)."

Response: We agree that providing both terms in this way improves clarity for the reader. We have revised the text accordingly to read: “Lyell's syndrome, also known as toxic epidermal necrolysis (TEN)” in the relevant section of the manuscript.

Additionally, the authors have included another figure in the manuscript (Figure 3), the authors believe that it will improve the quality of the work.

The authors included several references in the manuscript, which they also marked in red.

Thank you again for the valuable comments.

Reviewer 2 Report

Comments and Suggestions for Authors

This narrative review examines the dual role of lamotrigine as a first-line treatment for bipolar disorder and its association with severe cutaneous adverse drug reactions, particularly Stevens-Johnson Syndrome and Toxic Epidermal Necrolysis. The review covers pharmacology, clinical applications, and adverse effects of lamotrigine.

1. Figure 1 (mechanism of lamotrigine) is too simple, maybe the author could consider enhancing with more molecular detail.
2. Figure 2 is not clear.
3. Table 1 should be adjusted to three-line table.
4. Lack of PRISMA guidelines or othersystematic methodologies may introduce selection bias.
5. Although lamotrigine action in bipolar disorder is detailed, the link to Stevens-Johnson Syndrome pathophysiology such as sulfatase-mediated metabolite formation is not well described.
6. Why Stevens-Johnson Syndrome of lamotrigine risk is lower than carbamazepine/allopurinol despite shared HLA associations?

Author Response

Dear Reviewer,

Thank you for the opportunity to revise our manuscript entitled "Lamotrigine therapy: Relation between treatment of bipolar affective disorder and incidence of Stevens-Johnson Syndrome - A Narrative Review of the Existing Literature" [ID: jcm-3626988].

We appreciate the time and effort the reviewer have dedicated to evaluating our work and for their constructive feedback, which has helped us improve the quality and clarity of our manuscript.

Please find below our detailed responses to each of the reviewer’s comments. We have addressed all points raised to the best of our ability and have revised the manuscript accordingly. Changes made to the manuscript are highlighted in red.

Comment: “Figure 1 (mechanism of lamotrigine) is too simple, maybe the author could consider enhancing with more molecular detail.”

Response: We adjusted contents of Figure 1 to ensure it contains more information about molecular processes that are crucial for LTG action.

Comment: “Figure 2 is not clear.”

Response: The labels in the figure are now larger and more prominent, as they have been placed within rectangles with a solid monochromatic background.

Comment: “Table 1 should be adjusted to three-line table.”

Response: The table has been improved according to the reviewer's suggestions.

Comment: “Lack of PRISMA guidelines or other systematic methodologies may introduce selection bias.”

Response: We acknowledge the importance of standardized methodologies such as PRISMA in reducing selection bias in systematic reviews. However, the present manuscript was conceived and developed as a narrative review, aiming to provide a broad, critical overview of the current state of knowledge rather than to answer a narrowly defined research question through a systematic process. As such, the retrospective application of PRISMA guidelines is not feasible, given that the inclusion criteria, search strategy, and study selection process were not originally structured according to PRISMA protocol. Applying these guidelines post hoc would compromise the transparency and methodological integrity that PRISMA is designed to ensure.

Comment: “Although lamotrigine action in bipolar disorder is detailed, the link to Stevens-Johnson Syndrome pathophysiology such as sulfatase-mediated metabolite formation is not well described.”

Response: The importance of sulfated metabolites has been demonstrated in greater detail, with particular emphasis on their immunogenicity and the resulting T-cell responses.

Comment: “Why Stevens-Johnson Syndrome of lamotrigine risk is lower than carbamazepine/allopurinol despite shared HLA associations?”

Response: Lamotrigine and other compounds such as carbamazepine and allopurinol may cause Stevens-Johnson syndrome (SJS). The incidence of SJS with lamotrigine is estimated at 0.04%. Carbamazepine causes SJS/TEN in a frequency of about 14 per 100,000 users. Allopurinol when prescribed at dosages of 200 mg or above have  to increase the risk of SJS. The mechanisms of action of the compounds are complex and factors such as the structure of the drug and interaction with HLA molecules may contribute to the lower risk of SJS with lamotrigine compared to, for example, carbamazepine or allopurinol.

The mechanism of action of lamotrigine includes selective binding and inhibition of voltage-gated sodium channels, stabilization of presynaptic neuronal membranes, and inhibition of glutamate and aspartate release. It does not directly involve the same type of HLA interactions that are thought to be involved in the development of SJS with other drugs such as carbamazepine and allopurinol, among others. Their mechanism of action involves the formation of a haptenated peptide or noncovalent binding to HLA receptors or T cells.

Additionally, the authors have included another figure in the manuscript (Figure 3), the authors believe that it will improve the quality of the work.

The authors included several references in the manuscript, which they also marked in red.

Thank you again for the valuable comments.

Round 2

Reviewer 1 Report

Comments and Suggestions for Authors

Thank you to the authors for providing a revised version of the manuscript. While some concerns have been resolved, others remain unaddressed or only partially addressed.

In particular, the request "to include a broader range of genetic factors—both HLA and non-HLA—drawing on PharmGKB and other relevant sources", has not been adequately incorporated in the revision.

In fact, the four HLA variants currently listed in the manuscript on page 16 are inaccurate. HLA-B*15:11 and HLA-B*15:21 are associated with carbamazepine-induced SJS/TEN, not lamotrigine. They should be removed.

The statement, “Despite extensive research, we were unable to identify…” is surprising, given that well-documented associations between lamotrigine-induced SJS/TEN and specific genetic variants are available in the literature. For example, relevant HLA alleles for lamotrigine-induced SJS/TEN include HLA-A*02:07 [PMID: 29238301...], HLA-A*24:02 [PMID: 28351624; 28476759;…], and HLA-B*38:01 [PMID: 18192896; 27888155...]. On the non-HLA side, lamotrigine is primarily metabolized by UGT enzymes [PMID: 29238301; 33618992; …PharmGKB (https://www.pharmgkb.org/)], and variants in UGT genes are considered to affect the risk of severe cutaneous adverse reactions. These elements are essential for an accurate and complete description of the pharmacogenetics involved.

The section subtitle should be revised to: “Lamotrigine-induced SJS/TEN and Genetics.”

Author Response

Dear Editor,

Please find enclosed our revised manuscript entitled Response to Reviewer Comments – Manuscript “Lamotrigine therapy: Relation between treatment of bipolar affective disorder and incidence of Stevens-Johnson Syndrome - A Narrative Review of the Existing Literature” (by Kacper Żełabowski, Kacper Wojtysiak, Zuzanna Ratka, Kamil Biedka and Agnieszka Chłopaś-Konowałek) submitted for publication in Journal of Clinical Medicine.

Thank you very much for your reviews and the advice you have given us to make our work better. Below you will find manuscript corrections. All changes introduced to our initial manuscript have been marked up using the Changes color - red. As changes have been made to the text (parts of the text have been added/removed) the line numbering has changed, so that in the responses the authors refer to the new numbering, which is not consistent with that given by the reviewers.

The text was reread regarding English language and thus, minor revisions have been implemented into the manuscript.

Reviewer 2

Comment: “In particular, the request "to include a broader range of genetic factors—both HLA and non-HLA—drawing on PharmGKB and other relevant sources", has not been adequately incorporated in the revision.”

Response: In the subsection 2.2. Pharmacokinetics of Lamotrigine “It is extensively metabolized in the liver by the action of uridine diphosphate glucuronosyltransferases (UGT), mainly UGT1A4 and UGT2B7 isoforms to inactive metabolites 2-N-glucuronide conjugate, 5-N-glucuronide and a 2-N-methyl metabolite [27]”.

and 5.4. Lamotrigine-induced SJS/TEN and Genetics  there are fragments regarding the non-HLA UGT factor

„As mentioned in pharmacokinetics section (2.2. Pharmacokinetics of Lamotrigine) , UGT enzyme polymorphisms may be crucial for patient-to-patient differences in clinical response and toxicity and thus, SJS development. Notably, UGT1A4 and UGT2B7 isoforms are said to be involved, and drugs that interfere with UGT enzyme efficacy are able to modulate LTG's AUC, that is, exposure, and subsequently, risk for SJS development [27, 86]”.

Comment: “In fact, the four HLA variants currently listed in the manuscript on page 16 are inaccurate. HLA-B*15:11 and HLA-B*15:21 are associated with carbamazepine-induced SJS/TEN, not lamotrigine. They should be removed.”

Response: As recommended by the reviewer HLA-B*15:11 and HLA-B*15:21 were removed from the manuscript

Comment: “The statement, “Despite extensive research, we were unable to identify…” is surprising, given that well-documented associations between lamotrigine-induced SJS/TEN and specific genetic variants are available in the literature. For example, relevant HLA alleles for lamotrigine-induced SJS/TEN include HLA-A*02:07 [PMID: 29238301...], HLA-A*24:02 [PMID: 28351624; 28476759;…], and HLA-B*38:01 [PMID: 18192896; 27888155...]. On the non-HLA side, lamotrigine is primarily metabolized by UGT enzymes [PMID: 29238301; 33618992; …PharmGKB (https://www.pharmgkb.org/)], and variants in UGT genes are considered to affect the risk of severe cutaneous adverse reactions. These elements are essential for an accurate and complete description of the pharmacogenetics involved. “

Response: In subsection 5.4. Lamotrigine-induced SJS/TEN and Genetics, the authors of the manuscript inserted the following text fragment

“ A study by Koomdee et al. [117] found that HLA-A02:07 was present in 33.3% of LTG-induced CADR patients and showed significantly higher frequencies than both the treatment control and general population groups, although the sample was undeniably small (n=15), and the authors emphasized on the need for further research before a new screening marker in the for of 02:07 allele gets recognised as prognostic enough. Furthermore, in the subgroup analysis specifically for LTG-induced SCAR (which includes SJS), the association between HLA-A02:07 and SJS did not reach statistical significance, with the small sample being even more pronnounced here [117]. As mentioned in pharmacokinetics section (2.2. Pharmacokinetics of Lamotrigine), UGT enzyme polymorphisms may be crucial for patient-to-patient differences in clinical response and toxicity and thus, SJS development. Notably, UGT1A4 and UGT2B7 isoforms are said to be involved, and drugs that interfere with UGT enzyme efficacy are able to modulate LTG's AUC, that is, exposure, and subsequently, risk for SJS development [27, 86]. Typical agents being brought up are valproic acid and carbamazepine, phenytoin. In Flunitrazepam and clonazepam were concomitantly administered and such interaction was highly propable to take place, but nevetheless, sample consisting of 2 patients should not lead to generalized conclusions [117].

While not a major risk marker, HLA-B38:01 has been linked in some cases to in-creased susceptibility to SJS/TEN when triggered by certain medications. Exists an asso-ciation between HLA-B38:01 allele and an elevated likelihood of SJS/TEN triggered by certain antiepileptic drugs in the Spanish population. Carriers of that specific allele were more susceptible to developing SJS/TEN, with odds ratios ranging from approximatelly  13 to 147 depending on the control group [118]. By contrast, European study revealed limited correlation between HLA-B*38 allele and SJS/TEN development, observed in a small number of cases involving lamotrigine (19 cases) or sulfamethoxazole (28 cases). This suggests lower clinical significance of HLA-B38:01 allele, comparing to previously mentioned HLA-B58:01, where in the same study,  61% of affected individuals carried the allele and the odds ratio reached 80 [119]. Statistical correlation was identified between the HLA-A*24:02 allele and lamotrigine-induced SCAR in the southern Han population. Individuals who possessed this allele exhibited a more than fourfold increased likelihood of developing LTG-induced SJS/TEN (45.5% in those with the HLA-A*24:02 allele vs. 15.7% in those without the allele). HLA-A*24:02 has been identified as a significant risk factor for SJS/TEN in patients treated with the most commonly used aromatic ring antiepileptic drugs (lamotrigine, carbamazepine, phenytoin). The data presented by Yi-Wu Shi et al. served as the basis for the implementation of pretreatment screening in southern China [120]. A similar study was conducted in the Korean population. Koreans carrying HLA-A*24:02 were associated with an approximately 2-3-fold but statistically insignificant increase in the risk of LTG-induced SCAR (OR = 2.57; 95% CI 0.77-8.61; p = 0.14). This result highlights the need for studies on a larger cohort of Korean patients to unambiguously assess the association of the HLA-A*24:02 allele with cADR [121].

This fragment includes literature suggested by the reviewer, what is marked in red.

Comment: “The section subtitle should be revised to: “Lamotrigine-induced SJS/TEN and Genetics.”

Response: As recommended by the reviewer, the title of subsection 5.4 was changed from SJS and Genetic to Lamotrigine-induced SJS/TEN and Genetics.

Yours faithfully,

Agnieszka Chłopaś-Konowałek

Department of Forensic Medicine,

Division of Molecular Techniques,

Faculty of Medicine,

Wroclaw Medical University,

Sklodowskiej-Curie 52, 50369,

Wroclaw, Poland,

Correspondence: agnieszka.chlopas-konowalek@umw.edu.pl.; tel.: 781-654-694

Date 02.06.2025
